# A Highly Efficient Polystyrene-Based Cationic Resin to Reduce Bacterial Contaminations in Water

**DOI:** 10.3390/polym14214690

**Published:** 2022-11-03

**Authors:** Anna Maria Schito, Debora Caviglia, Gabriella Piatti, Silvana Alfei

**Affiliations:** 1Department of Surgical Sciences and Integrated Diagnostics (DISC), University of Genoa, Viale Benedetto XV, 6, 16132 Genova, Italy; 2Department of Pharmacy, University of Genoa, Viale Cembrano, 16148 Genoa, Italy

**Keywords:** cationic resin, swelling capacity, antibacterial properties, polymicrobial contamination, time–kill experiments, water decontamination

## Abstract

Nowadays, new water disinfection materials attract a lot of attention for their cost-saving and ease of application. Nevertheless, the poor durability of the matrices and the loss of physically incorporated or chemically attached antibacterial agents that can occur during water purification processes considerably limit their prolonged use. In this study, a polystyrene-based cationic resin (R4) with intrinsic broad-spectrum antibacterial effects was produced without needing to be enriched with additional antibacterial agents that could detach during use. Particularly, R4 was achieved by copolymerizing 4-ammonium-butyl-styrene (4-ABSTY) with N,N-dimethylacrylamide (DMAA) and using N-(2-acryloylamino-ethyl)-acrylamide (AAEA) as a cross-linker. The R4 obtained showed a spherical morphology, micro-dimensioned particles, high hydrophilicity, high-level porosity, and excellent swelling capabilities. Additionally, the swollen R4 to its maximum swelling capability, when dried with gentle heating for 3 h, released water following the Higuchi’s kinetics, thus returning to the original structure. In time–kill experiments on the clinical isolates of multidrug-resistant (MDR) pathogens of fecal origin, such as enterococci, Group B *Salmonella* species, and *Escherichia coli*, R4 showed rapid bactericidal effects on enterococci and *Salmonella*, and reduced *E. coli* viable cells by 99.8% after 4 h. When aqueous samples artificially infected by a mixture of the same bacteria of fecal origin were exposed for different times to R4 in a column, simulating a water purification system, 4 h of contact was sufficient for R4 to show the best bacterial killing efficiency of 99%. Overall, thanks to its physicochemical properties, killing efficiency, low costs of production, and scalability, R4 could become a cost-effective material for building systems to effectively reduce bacterial, even polymicrobial, water contamination.

## 1. Introduction

The contamination of water resources, including marine, lake, river, and drinking water, by pathogenic bacteria, such as *E. coli*, enterococci, and *Salmonella* spp., is one of the major current problems with a profound social and economic impact in the world [1]. The limited number of sanitation and water disinfection infrastructures means that over a billion people worldwide lack access to clean drinking water [1,2,3]. Drinking water, if contaminated with microorganisms, can lead to waterborne diseases such as giardiasis, cholera, gastroenteritis, and associated symptoms such as vomiting, abdominal cramps, and headache [4,5]. A lack of safe drinking water is estimated to cause around two million deaths each year, most of them children under the age of five [6]. To prevent the onset of waterborne diseases, there are effective water disinfection procedures, such as chlorination and ozonation [7,8]. However, these processes paradoxically lead to the formation of toxic disinfection byproducts, such as trihalomethanes and other carcinogens, which are harmful to human health and the environment [7,8].

In recent years, alternative technologies for water disinfection have been studied [9,10], also based on nanotechnologies. Antimicrobial nanomaterials such as silver nanoparticles (AgNPs) have attracted increasing attention for water purification. However, in addition to the data, albeit limited, on the possible danger of the absorption and nanotoxicity of NPs for both humans and the environment, the application of AgNPs for water purification is very expensive [11] and has been associated with mammalian cell cytotoxicity [12,13].

Very recently, antibacterial cellulose (cotton) nanocomposite fabrics (CNCFs) with in situ generated AgNPs using medicinal plant *Vitex* leaf extract have been prepared using a simple, low-cost, and environmentally friendly method [14]. In our opinion, although the scope of this study was to develop antibacterial cotton fabrics for medical applications, the materials produced by the authors could also have potential for application in water decontamination.

Recently, an attractive and environmentally friendly approach consisting of converting crayfish shells and their derivatives to functional materials for environmental applications, such as adsorption and/or antibacterial materials for waste-water purification, has been reported [15]. In this study, it was also demonstrated that the attachment of Ag-TiO_2_ nanoparticles to the crayfish-shell biochar carriers optimized their antibacterial ability and facilitated recycling [15]. The addition of effective antibacterial agents to infected waters could be an option to reduce bacterial contamination, but the difficult removal of residual antibiotics limits the application of this strategy. To overcome this problem and avoid post-treatment procedures, a smart solution could consist of immobilizing the antibacterial agents or inserting antibacterial compounds on insoluble polymeric matrices (resins) that act as a support [16]. With the birth of solid-phase organic synthesis (SPOS), various resins have been synthesized and employed in daily life for their many advantages, including excellent mechanical strength, high chemical stability, good weather resistance, and ease of further functionalization [17,18,19]. During studies to develop an alternative method of disinfection with insoluble polymeric materials, it was found that cross-linked poly-(vinyl-pyridinium halide) (PVPA) had a new and remarkable ability to remove bacteria from water. In this case, mechanistic studies revealed that PVPA was capable of irreversibly capturing bacteria during treatment [20].

Subsequently, ion-exchange resins have been used to retain microorganisms by direct treatment of cell suspensions [21,22,23,24]. Differently from the case previously reported, this time, the adsorption of microorganisms on the resins was reversible and the microorganisms were obtainable by desorption from the resins.

Additionally, polystyrene-based resins (PS resins), especially those crosslinked with divinylbenzene (i.e., P(St-DVB) resins), proved to be a good choice as a matrix substrate for disinfectant agents such as silver [25], quaternary ammonium salts [26], N-halamine [27], chitosan [28], antimicrobial peptides [29], etc. In this way, materials for the purification of drinking water can be obtained, thanks to the low cost of P(St-DVB) resins, their good mechanical strength, high porosity, and absorbency [25,26,27,28,29,30].

In this context, we recently reported the broad-spectrum bactericidal potency of a water-soluble cationic polystyrene-based copolymer (P5) obtained by copolymerizing the 4-ammonium butyl styrene hydrochloride salt (4-ABSTY), synthetized by us and chosen as an active monomer, with di-methyl-acrylamide (DMAA), chosen as the uncharged co-monomer [31].

On these results, aiming to develop a resin intrinsically capable of efficiently reducing the bacterial load in water, without the need for additional antibacterial agents losable during the purification processes, in this study, we synthesized the cationic cross-linked copolymer, namely resin R4, analogous to the water-soluble P5. This was accomplished by a one-step procedure, using the reverse-phase suspension copolymerization technique. In particular, the 4-ABSTY hydrochloride salt (M4 in this work, Figure 1) previously used as an active cationic monomer to obtain the water-soluble copolymer P5 was re-synthesized and copolymerized with DMAA in the presence of AAEA as a cross-linker under opportune conditions, later described, obtaining R4.

R4 was fractionated by sieving to achieve a central body of particles (>96% of the total amount subjected to sieving) with little dispersed granulometry, and very homogenous porosity, and first characterized by attenuated-total-reflectance (ATR) Fourier transformed infrared (FTIR) spectroscopy, to assess the presence of all predicted functional groups. Optical microscopy examinations were carried out both on R4, as obtained from polymerization, and on the swollen resin obtained after swelling it to its maximum swelling capability to evaluate its morphology and particle size in both cases, and to estimate its hydrophilicity and swelling capability in water. The water loss and swelling rate over time experiments were also conducted and reported to obtain the water release and absorption profiles. The determination of the maximum swelling capability was also used to confirm its high capacity to absorb water and determine its porosity. The content of NH_3_^+^ groups, essential to allow electrostatic interactions with the negatively charged surface of the bacteria and to capture, inhibit, and/or kill them, was determined following the method of Gaur and Gupta and confirmed using the results obtained by potentiometric titrations. The antibacterial properties of R4 were assessed first by carrying out classical time–kill experiments, exposing several clinical MDR isolates of fecal origin, including strains of *E. coli*, Group B *Salmonella* species, and species of the genus *Enterococcus*, to R4 for 4 h. Then, the same experiments were conducted on a mixture of strains from each of the previous species to investigate the potential negative or positive impact on the antibacterial efficiency of R4 of the co-existence of more bacterial species. Finally, the optimal contact time between aqueous samples containing a mixture of the same bacteria, which simulated a real contaminated water, and R4 in a hypothetical column disinfection system to have the best decontaminating activity of R4 was assessed.

## 2. Materials and Methods

### 2.1. Chemicals and Instruments

All reagents and solvents were from Merk (formerly Sigma-Aldrich, Saint Louis, MO, USA) and were purified by standard procedures. AAEA was prepared by known procedures [32] while M4 was prepared as previously described [31]. Organic solutions were dried over anhydrous magnesium sulphate and evaporated using a rotatory evaporator operating at a reduced pressure of about 10–20 mmHg. The melting ranges of solid compounds in this study were determined on a 360 D melting point device, resolution 0.1 °C (MICROTECH S.R.L., Pozzuoli, Naples, Italy). Melting points and boiling points are uncorrected. The FTIR spectra were recorded as films or KBr pellets on a Perkin Elmer System 2000 instrument (PerkinElmer, Inc., Waltham, MA, USA) while ATR-FTIR analyses were carried out using a Spectrum Two FT-IR Spectrometer (PerkinElmer, Inc., Waltham, MA, USA). ^1^H and ^13^C NMR spectra were acquired on a Bruker DPX spectrometer (Bruker Italia S.r.l., Milan, Italy) at 300 and 75.5 MHz, respectively. Mass spectra were obtained with a GC-MS Ion Trap Varian Saturn 2000 instrument (Varian, Inc., Palo Alto, CA, USA; EI or CI mode; filament current: 10 mA) equipped with a DB-5MS (J&W) capillary column. Elemental analyses were performed with an EA1110 Elemental Analyser (Fison Instruments Ltd., Farnborough, Hampshire, UK). UV-Vis spectra were acquired using a UV-Vis spectrophotometer (HP 8453, Hewlett Packard, Palo Alto, CA, USA) equipped with a 3 mL cuvette. HPLC analyses were performed on a Jasco model PU-980 instrument (JASCO Corporation, Hachioji, Tokyo, Japan), equipped with a Jasco Model UV-970/975 intelligent UV/Vis detector (JASCO Corporation, Hachioji, Tokyo, Japan) at room temperature. A constant flow rate (1 mL/min), UV detection at 254 nm, a 25 cm × 0.46 cm Hypersil ODS 5 mm column, and a mixture acetonitrile/water 6/4 as eluent were employed for the acquisitions. GC-FID analyses were performed on a Perkin Elmer Autosystem (Varian, Inc., Palo Alto, CA, USA), using a DB-5, 30 m, diameter 0.32 mm, film 1 mm capillary column. Column chromatography was performed on Merck silica gel (70–230 mesh). Thin layer chromatography (TLC) was carried out using aluminum-backed silica gel plates (Merck DC-Alufolien Kieselgel 60 F254, Merck, Washington, DC, USA), and the detection of spots was made by UV light (254 nm), using a Handheld UV Lamp, LW/SW, 6 W, UVGL-58 (Science Company^®^, Lakewood, CO, USA). Optical microphotographs were obtained with a Zeiss Axioskop instrument. Sieving was performed with a 2000 Basic Analytical Sieve Shaker-Retsch apparatus.

Potentiometric titrations were carried out using a Hanna Micro-processor Bench pH Meter (Hanna Instruments Italia srl, Ronchi di Villafranca Padovana, Padova, Italy), which was calibrated using standard solutions at pH = 4, 7, and 10 before titrations. Lyophilization was performed using a freeze–dry system (Labconco, Kansas City, MI, USA). Centrifugations were performed on an ALC 4236-V1D centrifuge at 3400–3500 rpm.

### 2.2. Synthesis of R4

A mixture of hexane (360 mL) and 205 mL carbon tetrachloride (CCl_4_) was placed in a round-bottom cylindrical flanged reactor equipped with an anchor-type mechanical stirrer and nitrogen inlet, thermos-stated at 35 ± 0.05 °C and deoxygenated by nitrogen (N_2_) bubbling for 30 min. Meanwhile, a solution was obtained by dissolving in a tailed test-tube under N_2_ the monomer M4 (4.08 g, 0.138 mol), DMAA (12.3 g, 0.797 mol), AAEA (1.72 g, 0.065 mol), and ammonium persulfate (APS) (0.34 g, 0.0015 mol, 1% wt/wt with respect to M4) in deoxygenated water distilled over KMnO_4_. Then, it was siphoned into the reaction vessel, and the density of the organic phase was adjusted by adding CCl_4_ so that the aqueous phase sank slowly when the stirring was stopped. The mechanical stirring was settled at 900 rpm, SPAN 85 dissolved in hexane (1 mL) was added to the mixture, and the polymerization was started. After 10 min, N,N,N,N,-tetra-methyl-ethylene-diamine (TMEDA) (680 µL) was introduced and the polymerization was continued for 90 min. The resin was filtered, washed with a series of selected solvents (in order 2-propanol, chloroform, water, absolute ethanol, chloroform, 2-propanol, and acetone), and then dried at reduced pressure and room temperature for 16–20 h to achieve a constant weight.

R4. White beads: 17.84 g, conversion yield 98%. ATR-FTIR (ν, cm^−1^): 3500–3000 (NH_3_^+^), 2926 (CH_2_), 1614 (C=O), 838 (CH bending phenyl 1,4-disubstituted).

### 2.3. Sieving of R4

The dried resin, after light grinding with a pestle to burst the fragile aggregates, was sieved by sieves with an external diameter (Ø_E_ = 10 cm) and meshes (25–120 mesh) chosen after preliminary tests on a small sample using the Analytical Sieve Shaker-Retsch 2000 Basic instrument (Retsch GmbH, Haan, Germany) as a vibrating base.

### 2.4. Maximum Swelling Capability (%) and Porosity (%) of R4

In a graduated centrifuge tube (Ø_est_ = 14 mm), a volume of R4 equal to 0.65 mL (*Vi*), corresponding to an amount of 188.7 mg (*Wi*), was introduced to which excess water was gradually added (10 mL). The gel obtained was gently shaken with a spatula for a few minutes to push away the trapped air and then sonicated for 15 min and degassed for an additional 9 min at room temperature using an Ultrasonic Cleaner 220 V (VWR, Milan, Italy). Upon centrifugation at 4000 rpm for 30 min, the volume and weight of the swollen resin was measured (*Vf* and *Wf,* respectively). The values of *Vi*, *Wi* and *Vf*, *Wf* were used to calculate the percentage porosity (*P*%) and the maximum percentage swelling capability (*S*%) according to the following equations: Equations (1)–(4):(1)$$P\;\left(\%\right)=\frac{Vf-Vi}{Vf}\;\times \;100$$
(2)$$P\;\left(\%\right)=\frac{Wf-Wi}{Wf}\;\times \;100$$
(3)$$S\;\left(\%\right)=\frac{Vf-Vi}{Vi}\;\times \;100$$
(4)$$S\;\left(\%\right)=\frac{Wf-Wi}{Wi}\;\times \;100$$

### 2.5. Weight Loss (Water Loss)

An exactly weighed sample of the swollen R4 (263.7 mg), obtained in the previous experiment, was deposited in a Petri dish (PD). PD was then placed in an oven under a controlled temperature (37 °C), and the weight loss was monitored as a function of time until a constant weight was reached. The cumulative *weight loss* percentage was determined by means of Equation (5):(5)$$Weight\;loss\;\left(\%\right)=\frac{MQ-Mt}{MQ}\;\times \;100$$
where *MQ* and *Mt* are the initial mass of the swollen resin and its mass after a time *t*, respectively.

### 2.6. Equilibrium Swelling Rate

The swelling measurements were carried out by immersing 10.1 mg of the dried resin obtained in the previous experiment in deionized water in a test tube. At time intervals of 15 min, the sample in the test tube was centrifugated (15 min, 4000 rpm) to remove the non-absorbed water, inverted on filter paper to absorb residual water, and weighed. The cumulative swelling ratio percentage (*Q*%) as function of time was calculated from Equation (6):(6)$$Q\;\left(\%\right)=\frac{Ws-Wd}{Wd}\;\times \;100$$
where *Wd* and *Ws* are the weights of the dried and swollen resin, respectively. The equilibrium swelling ratio (*Q*_equil_) was determined at the point the hydrated resin achieved a constant weight.

### 2.7. Estimation of NH_3_^+^ Content in R4

The NH_3_^+^ equivalents contained in R4 were estimated following the method of Gaur and Gupta [33]. Briefly, R4 (about 2 mg) was taken in a 2 mL Pierce reaction vial to which 4-O-(4,40-dimethoxytriphenylmethyl)-butyryl (0.25 mL) (reagent A), a catalytic amount of di-methylamino-pyridine (DMAP) (5 mg), and triethylamine (TEA) (100 µL) were added. The vial was screw-capped and gently tumbled at room temperature for 30 min. The reaction mixture was transferred to a 2 mL sintered funnel and washed successively with N,N-dimethylformamide (DMF) (2 × 10 mL), MeOH (2 × 10 mL), and, finally, dry diethyl ether (2 × 10 mL). After the polymer support was dried under vacuum, a weighed quantity (2.1 mg) was taken in a 10 mL volumetric flask, which was then filled up to the mark with HClO_4_ (reagent B). The released 4,40-dimethoxytriphenylmethyl cation (ε = 70,000) was estimated spectrophotometrically at 498 nm against reagent B as the blank.

### 2.8. Potentiometric Titration of R4

Potentiometric titrations were performed at room temperature and the titration curves of R4 were obtained. Particularly, exact amounts of R4 (36.8 mg) were suspended in 50 mL of Milli-Q water (m-Q) and treated under magnetic stirring with a standard 0.1 N NaOH aqueous solution (3.0 mL, pH = 10.41). The solutions were potentiometrically titrated under stirring by the addition of 0.5 mL, up to 6.0 mL, and finally 1.0 mL up to a total of 9.0 mL and the pH values of the obtained solutions were measured. Titrations were made in triplicate and the measurements were reported as the mean of three independent experiments ± SD.

### 2.9. Microbiology

#### 2.9.1. Microorganisms

A total of 14 strains of different species of Gram-positive and Gram-negative fecal bacteria implicated in the water contamination were utilized to assess the antibacterial effects of R4 in this study. All were clinical isolates from human specimens and were identified by the matrix-assisted laser desorption/ionization time-of-flight (MALDI-TOF) mass spectrometric technique (Biomerieux, Firenze, Italy) or by VITEK^®^ 2 (Biomerieux, Firenze, Italy). Four Gram-positive organisms were used, including two strains belonging to the *Enterococcus* genus (two *Enterococcus faecalis* resistant to vancomycin (VRE) and two *E. faecium* VRE). The remaining 10 isolates were *Enterobacteriaceae* and included 5 *E. coli* isolates (of which one was a New Delhi metallo β-lactamase (NDM) producer isolate, one was a fully susceptible strain, and 3 were carbapanemase-producing isolates) while the other 5 strains were Group B *Salmonella* species.

#### 2.9.2. Time–Kill Experiments

Time–kill (TK) experiments were performed with R4 on 4 VRE isolates of the *Enterococcus* genus, 5 isolates of *E. coli*, and 5 isolates of the Group B *Salmonella* species, using a previously reported procedure [34], slightly modified, according to indications in the literature [35], and because of the physicochemical features of R4, which is an insoluble support. In addition, to mimic a real fecal contamination, TK experiments with R4 were conducted on a mixture of the same bacterial species previously tested individually. Briefly, a mid-logarithmic phase culture was diluted in PBS (10 mL) containing 25 mg/mL of R4 to obtain a final inoculum of approximately 5 × 10^5^ CFU/mL. The same inoculum was added to PBS without R4 as a growth control. Tubes were incubated at 37 °C with constant shaking. Samples of 0.20 mL from each tube were removed at 0, 2, and 4 h; appropriately diluted with a 0.9% sodium chloride solution to avoid the carryover of R4 being tested; plated onto Mueller–Hinton (MH) broth (Merck, Darmstadt, Germany) plates; and incubated for 24 h at 37 °C. Growth controls were run in parallel. The percentage of surviving bacterial cells was determined for each sampling time by comparing the colony counts with those of standard dilutions of the growth control. TK experiments were performed over 4 h and repeated until a reproducibly was achieved. The results expressed as log_10_ of viable cell numbers (CFU/mL) were reported in a dispersion graph vs. time values (0, 2, 4 h), obtaining the time–kill curves. The time–kill curves shown in Section 3.9.1 of this article are those obtained for a representative strain for each species, and particularly those whose trend was observed most frequently for each isolate.

Furthermore, the results associated with the obtained time–kill curves were expressed both as a log_10_ reduction (RLog_10_ (CFU/mL)) in the original cell number and as a percentage of the killing efficiency (K (%)) of R4 over time. R4 was retained bactericidal when it caused a ≥3 log_10_ decrease in CFU/mL (≥99.9% killing) in the initial inoculum [36].

#### 2.9.3. Determination of the Optimal Contact Time between Artificially Contaminated Water and R4

Experiments to determine the optimal contact time between contaminated waters and R4 in a hypothetical disinfection system to obtain the best decontaminant activity were carried out in a small column of glass (Ø = 1 cm, h = 10 cm) equipped with a tap and filled with 25 mg of R4. Samples of artificially contaminated water were prepared first from a single strain of *E. coli* and secondly, more realistically, from a mixture of the same fecal bacteria used in previous experiments. Particularly, samples were prepared by diluting a mid-logarithmic phase culture of strains rejuvenated for 2 h in PBS (10 mL) to obtain a final inoculum of about 1.0 × 10^5^ CFU/mL. A volume of 5 mL of the prepared inoculum was placed in the column and left in contact with R4 at room temperature for 2, 4, and 24 h in the explorative case using *E. coli* while a maximum of 4 h in the realistic case using the bacterial mixture, before proceeding to a rapid filtration under vacuum. Samples of 0.20 mL of each filtered volume were appropriately diluted with a 0.9% sodium chloride solution to avoid the carryover of R4 being tested, plated onto MH plates, and incubated for 24 h at 37 °C. Growth controls were run in parallel. The percentage of surviving bacterial cells was determined for each sampling time by comparing the colony counts with those of standard dilutions of the growth control. Experiments were repeated until a reproducibly was achieved.

The results were expressed as Log_10_ reduction (dLog_10_ (CFU/mL)) in the original cell number and as the killing percentage efficiency of R4 (K (%)) as a function of the contact times, and the reported values were those that were observed most frequently (modal values).

## 3. Results and Discussion

The antibacterial mechanism for several ammonium salts, both in the form of low molecular weight and soluble compounds [37,38,39], and in the form of macromolecular soluble ammonium salts (polymers, copolymers, and dendrimers) [31,38,39,40,41,42] has long been studied. The antibacterial activity of these cationic compounds is due to their ability to kill bacteria thanks to a sequential series of elementary processes. In particular, ammonium salts are first adsorbed on the bacterial cell surface by electrostatic interactions [38,39] and then spread through the cell wall. As a result of their binding to the cytoplasmic membrane (CM), they significantly damage its integrity and cause the release of cytoplasmic constituents, such as K^+^ ions, DNA, and RNA, thus determining the death of the bacterial cell [38,39]. The antibacterial activity of soluble polymeric phosphonium salts of various structures, as a function of their positive charge density and the length of their chains, has been thoroughly studied. The results showed that precisely the positive charge density (responsible for hydrophilicity and greater solubility in water) and the length of their chains (responsible for the hydrophobic character) represent the key points of the biocidal capacity of cationic macromolecules. It has been shown that the high density of positive charge improves the ability of such molecules to interact with the negatively charged bacterial surface while long hydrophobic chains promote diffusion and strong interaction with CM, thus determining a higher antibacterial/bactericidal activity [38,39]. Collectively, a correct balance between the hydrophilic and hydrophobic properties is desirable to have a remarkable antibacterial effect. To this end, to obtain a resin with intrinsic powerful bactericidal effects, we strategically used the same M4 cationic monomer that supplied the soluble copolymer P5 [31], endowed with powerful bactericidal activity and therefore an optimal hydrophilic-lipophilic balance (HLB). In this case, to obtain an insoluble resin, we copolymerized M4 with the same co-monomer used to prepare P5 but in the presence of a suitably selected crosslinker. Although, in the literature, it is suggested that polystyrene-based resins (PS resins) crosslinked with divinylbenzene (i.e., P(St-DVB) resins) may be a good choice as a matrix substrate for disinfection agents, including ammonium salts [26], to obtain a low-cost material to purify water with good mechanical strength, high porosity, and absorbency [25,26,27,28,29,30], we cross-linked M4 and DMAA with AAEA. Hydrophilic AAEA was used instead of hydrophobic DVB to avoid impairing HLB of M4 and obtain a material that, although insoluble, would be able to interact well with water, by swelling, and therefore with any bacteria present.

### 3.1. Synthesis of R4

The M4 monomer was converted into resin R4 by reverse suspension polymerization at 35 °C. Particularly, according to slightly modified reported procedures [43,44], monomer M4, co-monomer DMAA, and the cross-linker AAEA previously dissolved in water were suspended in a mixture of CCl_4_/hexane. SPAN 85 and APS/TMEDA were used as the anti-coagulant and initiator, respectively (Figure 1).

The polymerization was repeated several times and carried out in cylindrical equipment such as those shown in Figure 2, depending on the total volumes needed. The procedure described in Section 2.2 was that of an optimized representative reaction.

In fact, it is known that containers with cylindrical geometry are able to minimize the horizontal component of the stirring motion of the suspension, responsible for the tendency of the micro-drops to aggregate [45]. After 90 min, the polymerization was interrupted and the precipitated R4 was separated by filtration, and then washed repeatedly with successively isopropanol, chloroform, water, ethanol, and acetone. R4 was then vacuum-dried to constant weight and stored at room temperature for subsequent sifting operations. The conditions applied afforded very good conversions (98%).

### 3.2. ATR-FTIR Spectra

The ATR-FTIR analyses were made directly on samples of R4, as obtained by the polymerization reaction and on a sample of R4 achieved after drying a sample of R4 swollen at its maximum swelling capability. Spectra were acquired in triplicate and those reported in Figure 3 are representative images.

As expected, the ATR-FTIR spectrum of R4 showed a weak band of CH stretching of methylene groups (2926 cm^−1^) deriving from the monomer M4, and a strong band at 1614 cm^−1^, typical of dimethyl acrylamide derivatives, and in this case belonging both to the co-monomer and the crosslinkers, thus assessing the presence of all three main ingredients in the structure of R4. No bands in the range 900–911 cm^−1^ were observable, thus establishing the absence of residual monomers. The pink spectrum given by dried R4 obtained by the experiment of weight loss described in Section 2.5 appeared to be identical to that given by pristine R4, thus establishing that the interactions that occurred between water and R4 during swelling are fully reversible.

### 3.3. Sieving of R4 and Optical Microscopy

R4 was fractioned by sieving using sieves with 25–120 mesh, obtaining beads with the bulk of the material >96% of the original weight. The microstructure of R4 of these particles was investigated by acquiring optical microphotographs with a Zeiss Axioskop instrument on the sieved resin. In the optical images of the sieved dried R4, particles appeared as well-separated microspherular beads with the bulk of the material in the size range 125–500 µm (Figure 4).

Regarding this, the reaction vessel/stirrer combination adopted here was particularly convenient for the preparation of well-separated crosslinked polystyrene-based resin beads with well-defined average particle sizes, such as those of R4 shown in Figure 4 [45]. It has been reported that other factors probably affected the average particle size and distribution during the suspension polymerization, including the type of suspension stabilizer used (SPAN 85), diameter of the vessel and stirrer, stirrer speed, and monomer-to-water ratio [45].

### 3.4. Maximum Swelling Capability (%) and Porosity (%) of R4

The maximum swelling capability percentage of R4 by volume and weight was determined after 15 min of sonication and 9 min of degassing of a fine dispersion of R4 (0.65 mL, 188.7 mg) in an excess of water using Equations (3) and (4). The data obtained by such experiments were useful for finding the porosity ratio percentage using Equations (1) and (2). The determinations were made in triplicate and the results are reported in Table 1, expressed as the mean of three independent experiments ± standard deviation (SD). Appendix A associated with this paper shows the aspect of swollen R4 obtained from hydrating the dry resin with an excess of 10 mL of water and recovered by centrifugation at 4000 rpm for 30 min while Figure 5 shows a representative optical microphotograph of the swollen resin.

As reported in Table 1, R4 was capable of absorbing so much water that it increased its volume by 731% and its weight by 2444%, respectively. The porosity was 88% and 96%, thus confirming both a remarkable porosity and absorbing ability. The optical microphotograph of the swollen R4 (Figure 5) substantiated the high absorbing capability of R4, showing pseudo-spherical beads unequivocally imbibed with water, with sizes up to 1300 µM. This meant that swollen R4 showed beads up to 10.4-fold greater than the smallest dried particles (1040%), thus cross-validating the data reported in Table 1. These characteristics, in a possible use of R4 as a material to reduce bacterial contamination in water, should favor electrostatic interactions with the bacteria contained in it. In addition, the same characteristics of high porosity and hydrophilicity indicate the great compatibility of R4 with water and good flow properties, which, in a potential use as a material to build columns for water disinfections, could allow an easy and rapid flux of contaminated waters.

### 3.5. Weight Loss (Water Loss)

Figure 6 shows the appearance of the soaked R4 (SR4) obtained in the previous experiment (a) and the fully dried R4 (DR4) obtained by heating SR4 at 37 °C for 6.5 h.

As observable in Figure 6, when heat-dried, R4 leaves a porous amorphous solid.

Appendix A shows the data of the experiments, including the weights of R4 determined at times T_0_–T_7_ and the related cumulative weight loss percentages while Figure 7a shows the cumulative weight loss percentage curve of SR4 over time and gentle heating. Figure 7b shows, in the same color, the Higuchi kinetic model, which is the mathematical model that best fit the data of the cumulative weight loss curve in Figure 7a.

As observable in Figure 7, the weight loss was quantitative, and the equilibrium was already reached after three hours. To identify the exact kinetics and the main mechanisms that governed the loss of water from the swollen R4, we fit the data of the curve in Figure 7 with the zero-order model (% cumulative water release vs. time), first-order model (Log_10_ % cumulative water remaining vs. time), Hixson–Crowell model (cube root of % cumulative water remaining vs. time), Higuchi model (% cumulative water release vs. square root of time), Korsmeyer–Peppas model (Ln% cumulative water release vs. Ln of time), and Weibull model (LnLn(100/residuals (%) vs. Ln of time) to obtain the related dispersion graphs [46,47,48,49,50,51]. The highest value of the coefficient of determination (R^2^) of the equations of the linear regressions of such graphs was considered as the parameter to determine which model better fit the water release data. The R^2^ values are reported in Table 2 and indicate that the water loss from SR4 best fit the Higuchi kinetic model (Figure 7b).

According to what is reported in the literature [51], the Higuchi equation can be represented in the simplified form reported here and Equation (7):Q = KH × t^1/2^(7)
where Q is the cumulative release of water by DR4, t is the time, and KH is the Higuchi release rate constant. Therefore, the simple Higuchi model results in a linear Q versus t^1/2^ plot having a gradient, or slope, equal to KH (in our case, equal to 48.8) and we say the system follows t^1/2^ kinetics. Hence, we can interpret that the prime mechanism of water release from SR4 is a Fickian diffusion-controlled release mechanism.

### 3.6. Equilibrium Swelling Ratio

The swelling measurements were carried out at fixed times following the procedure described in Section 2.6 until the weight of the swollen resin was approximately constant.

Appendix A collects the data of the experiments, including the weights of R4 at times T_0_–T_3_ and the cumulative swelling ratio percentages, while Figure 8 shows the cumulative swelling ratio percentage curves of R4.

As observed, the equilibrium swelling ratio (*Q*_equil_), which was determined at the point the hydrated gels achieved a constant weight, was already reached at T_1_ (15 min). In this case, the other determinations were used to obtain an average value ± SD, which is reported in Appendix A (last column).

### 3.7. Estimation of NH_3_^+^ Content in R4

The NH_3_^+^ content of the resins was estimated following the method of Gaur and Gupta, which, among others available in literature for determining the NH_3_^+^ group content in insoluble matrices, is signaled as being operator-friendly and sensitive [33]. Particularly, it is based on the labeling of the amino groups with 4-O-(4,40-dimethoxytriphenylmethyl)-butyryl residues and the quantitative determination through UV-Vis spectroscopy of the 4,40-dimethoxytriphenylmethyl cation (ε = 70,000 at 498 nm) released from the resin after treatment with HClO_4_. From the values of the absorbance (A) determined at 498 nm, the NH_3_^+^ moles for 1 g of R4 were estimated using Equation (8):(8)$$\frac{MolesNH2}{R4\left(g\right)}=\frac{A\times V}{p\times 70,000}$$
where *V* is the volume of the mixture used for detritylations and *p* is the weight in milligrams of the resin functionalized and subjected to detritylations. The results, expressed as the mean of independent determinations ± SD, are included in Table 3.

### 3.8. Potentiometric Titration of R4

The titration curves were obtained by graphing the measured pH values vs. the aliquots of HCl 0.1N added (Figure 9a). Subsequently, from the titration data, the dpH/dV values were computed and reported in the graph vs. those of the corresponding volumes of HCl 0.1N, thus obtaining the first derivative line of the titration curve (Figure 9b). The potentiometric titrations of R4 were allowed to titrate the NH_3_^+^ groups of R4 and further determine the NH_3_^+^ content in R4 already estimated by the method of Gaur and Gupta [33] in the previous section.

As observable in Figure 9a, the titration curve of R4 showed a significant buffer capacity and two very small jumps in the pH values when 3 and 5 mL of HCl 0.1 N were added while the end titration point was visible when 6 mL of HCl 0.1 N was reached. Particularly, by determining the first derivatives of the titration curves, whose maxima represent the titration end points (or the various phases of the protonation process), we observed two small maxima at pH = 8.33 and 6.33 indicating an early two-phase protonation while the highest maximum representing the final end point was visible at pH = 4.22. Table 4 collects the results related to the content of the NH_3_^+^ group in R4, which confirmed the values determined by UV-Vis determinations previously reported (Section 3.7), with a very small error of 1.15%.

### 3.9. Microbiology

#### 3.9.1. Time–Kill (TK) Experiments

Hypothesizing the possible use of R4 as a new material for the decontamination (by contact) of water infected by bacteria, we explored the possible bactericidal effects of R4 by carrying out TK assays in suspension. In particular, TK experiments were carried out for 4 h with R4 on selected fecal isolates, including MDR strains, potentially present in non-potable water as described in Section 2.9.2. The experiments were repeated until reproducibly was achieved. The Log_10_ (CFU/mL) of the surviving bacterial cells for a period of 4 h are reported in the dispersion graphs with respect to the time values (0, 1, 2, 4 h), thus obtaining the time–kill curves. The lines whose trend was observed most frequently are reported in Figure 10 and Figure 11. Figure 10 shows the most representative curves obtained for each species studied singularly (Figure 10a–d) while Figure 11 shows the results obtained when R4 was tested on a mixture of the same isolates reported in Figure 11. In particular, Figure 10a–d show the results obtained with the Group B *Salmonella* species strain 505, *E. coli* 462 (a strain producing New Delhi metallo-β-lactamases), *E. faecium* 503 (VRE), and *E. faecalis* 365 (VRE), respectively.

In both cases, the results were expressed as the Log_10_ reduction (RLog_10_ (CFU/mL)), and the killing percentage efficiency (K (%)). The values reported in Table 5 and Table 6 are those that were observed most frequently.

According to Figure 10a,b, R4 was bactericidal against Group B *Salmonella* species 505 after 4 h of exposure while it cannot be considered bactericidal against *E. coli* 462 since the Log_10_ reduction was <3 (Table 5). When TK experiments were performed on a mixture of the same strains that were previously tested singularly, the observed bactericidal effects of R4 on *Salmonella* 505 when tested alone were initially maintained, and a Log_10_ reduction = 3 in the initial inoculum of both *E. coli* 462 and *Salmonella* 505 in mixture was detected after 2 h of exposure (Figure 10, Table 6). Up to 4 h, even if the simultaneous presence of different bacterial species favored a certain bacterial regrowth of both *E. coli* and *Salmonella* (Figure 11 and Table 6), the reduction in the killing percentage efficiency of R4 was trivial since it changed from 99.99% to 99.8%.

Interestingly, against all strains of *Enterococcus* considered, R4 showed rapid bactericidal effects after only 2 h of exposure, and a killing percentage of 100% was observed over all times of exposure, both when VRE *E. faecium* 503 and *E. faecalis* 365, chosen as representative strains, were singularly tested and when they were in a mixture in the presence of *E. coli* 462 and Group B *Salmonella* species 505.

#### 3.9.2. Determination of the Optimal Contact Time of a Highly Contaminated Water Model with a Column Containing R4

In this part of the study on R4, we evaluated its potential environmental application as a new material for sanitation systems capable of reducing the fecal bacterial contamination of polluted water. In particular, a small amount of R4 was inserted in a glass column (Ø = 2.5 cm), up to a thickness of 3 cm, to imitate a hypothetical disinfection system working by contact followed by rapid filtration. Then, the best contact time between the aqueous models of contaminated water (artificially prepared) and R4, before filtration, for obtaining the best decontaminant activity was determined. Appendix A shows the column and the appearance of R4 in contact with the aqueous model of contaminated water (Appendix A) and just after filtration (Appendix A). In an early investigation, an aqueous model of water containing *E. coli* 224, a strain fully susceptible to antibiotics, was used to explore whether, after 4 h of exposure, bacterial regrowth could occur. According to the obtained results, over 4 h, regrowth occurred, and a decrease in the killing efficiency percentage from 97% (4 h) to 31% (24 h) was evidenced (Appendix A). After establishing this, in order to obtain the best efficient decontamination effects, it is not necessary to exceed contact times between R4 and bacteria of four hours. A more realistic aqueous model of infected water was prepared using MDR strains of *E. faecium* 503, *E. faecalis* 365, *E. coli* 462, and strain 505 of a *Salmonella* species of Group B. Therefore, experiments of the treatment of the contaminated aqueous model were carried out in the column containing R4, as previously described, and determinations of the Log_10_ (CFU/mL) after contact times of 1, 2, and 4 h before rapid filtrations were made. Determinations of Log_10_ (CFU/mL) were made in parallel on the same mixture of bacteria not treated with R4 at the same times of exposure (control (CTR). Results were reported both as the variation of Log_10_ (CFU/mL) (dLog_10_ (CFU/mL) and as the percentage of killed bacteria when the mixture was exposed to R4 for 1, 2, and 4 h (Table 7, Figure 12a,b). The results unequivocally demonstrated that the optimal contact time for obtaining a higher reduction in bacterial contamination was 4 h, thus confirming what was previously observed with *E. coli* 224.

Even when it was used in a column simulating a sanitization system acting by filtration, R4 proved less efficient in reducing the bacterial contamination and lost the bactericidal power demonstrated in the TK experiments against enterococci. It was unexpectedly more efficient in killing bacteria when used against the bacterial mixture than when it was used against a single species (*E. coli* 224) (Figure 12b).

Collectively, after 4 h of contact of the bacterial mixture with R4, followed by rapid filtration, the initial number of bacterial cells was reduced by 2.3 Log_10_, corresponding to 99.4% of the bacteria killed, while an increase of 2.25Log_10_ was observed for the control.

Recently, an iodine-doped pyrrolidone-based polymer poly-(N-vinyl-2-pyrrolidone-co-vinyl acetate) (P(VAc-NVP)-I) was used in a filtration system, made of three consecutive filtration columns containing it, for water purification experiments against water containing *E. coli* as a model for bacteria-contaminated water [52]. According to the reported results, differently from the control, when the contaminated water was passed through the P(VAc-NVP)-I-packed columns, *E. coli* was completely inactivated and the sterilization efficiency of P(VAc-NVP)-I in the columns was close to 100% [52], as in our case.

Although the reported results are very promising and interesting, the bactericidal effects of R4 in the column were remarkably more potent. In fact, instead of filtration through three consecutive filtering columns containing unknown quantities of P(VAc-NVP)-I, three consecutive filtrations (after 1, 2, and 4 h) through the same minimum quantity of R4 in the column were sufficient to obtain the same sterilization efficiency. In addition, in this study, we extended the evaluation of the sterilization efficiency of R4 to other relevant bacteria of fecal origin that can be found in contaminated water. So, unlike the study by Borjihan et al., where the sterilization efficiency of P(VAc-NVP)-I was demonstrated only in the presence of *E. coli*, R4 demonstrated a sterilization efficiency of over 99% even in the presence of a polymicrobial pollution by enterococci, *E. coli*, and group B *Salmonella* species. Additionally, the overall merit of our study consists in having tested R4 on several clinical MDR isolates of the selected species.

## 4. Conclusions

In this study, a high-potential polystyrene-based cationic resin (R4) was developed and tested for use as a new low-cost production material in sanitization systems to reduce bacterial contamination in water. In particular, it was fabricated by a simple one-step reaction by copolymerizing 4-ABSTY with DMAA and using AAEA as a cross-linker. Since R4 has intrinsic broad-spectrum antibacterial effects without needing to be enriched with additional antibacterial agents, it does not suffer from a typical problem of most materials used for water decontamination, which are likely to release the included antibacterial molecules during use, thus losing their antibacterial efficiency over time. The presence of all ingredients (4-ABSTY, DMAA, and AAEA) in R4 was confirmed by ATR-FTIR spectroscopy. The fractionation by sieving of R4 provided a central body of particles (>96% of the total quantity sieved) with a little dispersed granulometry. R4 was characterized by several other analytical techniques, which showed a spherical morphology and micro-dimensioned particles (microspheres), endowed with high hydrophilicity, large swelling capability in water, very high and homogeneous porosity, and the capability to return to the original structure after swelling. The content of NH_3_^+^ groups, essential for allowing electrostatic interactions with the negatively charged surface of bacteria and to capture, inhibit, and/or kill them, was determined both by following the method of Gaur and Gupta and by the results from potentiometric titrations. The antibacterial properties of R4 were first assessed by performing time–kill experiments up to 4 h of exposure on MDR clinical isolates of *E. coli*, *S. aureus*, and *Salmonella* and *Enterococcus*. Subsequently, similar experiments were conducted by simulating a hypothetical disinfection system (consisting of a column containing R4 and using a model of water that was artificially contaminated with fecal species such as *E. coli*, Group B *Salmonella* species, *E. faecium*, and *E. faecalis*). These experiments established that the optimal contact time between R4 and water for achieving the death of more than 99% of bacteria is 4 h.

## Data Availability

All data referred to this study are included in the present manuscript.

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
