# Peer review of "A Highly Efficient Polystyrene-Based Cationic Resin to Reduce Bacterial Contaminations in Water"

_polymers, 2022, doi:10.3390/polym14214690_

Round 1

Reviewer 1 Report

Schito et al. reported A Highly Efficient Polystyrene-Based Cationic Resin to Reduce 2 Bacterial Contaminations in Water. The manuscript have sufficient novelty and originality. The manuscript need a revision before publication. The graphs are not in proper format, make again. Optical Microscopy results need to explain in details otherwise remove these images. Their is few typo mistake in the manuscript, need to improve the language of manuscript.

Author Response

Schito et al. reported A Highly Efficient Polystyrene-Based Cationic Resin to Reduce Bacterial Contaminations in Water. The manuscript have sufficient novelty and originality. The manuscript need a revision before publication. The graphs are not in proper format, make again.

On suggestion of the Reviewer, we have double-checked all the graphs in our manuscript, which were obtained by the most used software, including Chemdraw Ultra 7.0 for drawing chemical structures and lab. glass-made instruments, Microsoft excel, to obtain dispersion graphs, bar graphs, graphs of cell viability vs. times or concentrations, linear trend lines etc., SpectraGryph 1.2 spectroscopy software, to visualize and elaborate ATR-FTIR spectra. Sincerely, we found them in a proper format. We also kindly point out to the Reviewer that we have already published many articles in Polymers or other MDPI journals, containing graphs in analogous presentations, which have been always considered in the proper format, both by Reviewers and the Editorial Office. We kindly ask the Reviewer to be satisfied with our graphs.

Optical Microscopy results need to explain in details otherwise remove these images.

Optical micrographs have been explained in detail, as asked. Please, see lines 396-402 and 420-424.

Their is few typo mistake in the manuscript, need to improve the language of manuscript.

As suggested by the Reviewer, the whole manuscript has been double-checked with the help of Professor Deirdre Kantz, native English teacher, who currently works for the University of Genoa and Pavia, to find and correct typos and improve the language quality of our work.

Reviewer 2 Report

Water disinfection materials attract a lot of attention for their cost-saving and ease of application. The manuscript is well organized and interesting. However, there are some issues need to be fixed. Major revision is suggested and the comments are listed as below.

1.     There are too many keywords and the keywords are too long. Usually, 5-6 keywords are enough.

2.     A wide variety of approaches and technologies have been developed for the detection and monitoring of various bacteria. Some closely related references are suggested to be cited in the introduction for broad readers, for example “A review on conversion of crayfish-shell derivatives to functional materials and their environmental applications; Preparation and properties of cellulose nanocomposite fabrics with in situ generated silver nanoparticles by bioreduction method”.

3.     Please pay attention of the writing of degree Celsius “oC”.

4.     Is “NH3+” in line 214 right? NH4+?

5.     “After 90 min.” in line 350 should be revised as “After 90 min,”. “.” is missing for the sentence “The conditions applied afforded very good conversions (98%)” in line 354. Please double check the whole manuscript to remove such typos.

6.     The scale bar in Figure 5 is hard to be recognized. Please replace it with a clear one.

7.     Please redraw the fitting curve in Figure 7b. The data before 1.75 h show good linearship.

8.     It would be better to do some comparison on the bactericidal performances with other polymers. More references published recently are suggested to be cited. Please double check the writing of reference. Some journal names are written in full name while others are not.

Author Response

Water disinfection materials attract a lot of attention for their cost-saving and ease of application. The manuscript is well organized and interesting. However, there are some issues need to be fixed. Major revision is suggested and the comments are listed as below.

  1. There are too many keywords and the keywords are too long. Usually, 5-6 keywords are enough.

We thank the Reviewer for his suggestion. So, even if Polymers allows up to ten keywords, we have shortened the too long keywords and reduced them to six. Please see lines 29-32.

  1. A wide variety of approaches and technologies have been developed for the detection and monitoring of various bacteria. Some closely related references are suggested to be cited in the introduction for broad readers, for example “A review on conversion of crayfish-shell derivatives to functional materials and their environmental applications; Preparation and properties of cellulose nanocomposite fabrics with in situ generated silver nanoparticles by bioreduction method”.

We thank the Reviewer for his suggestion. Accordingly, the two references (new Ref. 14 and 15) suggested by the Reviewer have been cited and discussed in the Introduction Section (lines 55-66) and included in the references list.

  1. Please pay attention of the writing of degree Celsius “oC”.

We apologise in advance with the Reviewer, but we have carefully checked all the manuscript (version word uploaded on Polymers) and the degree Celsius is correctly written along all work (i.e. °C).

  1. Is “NH3+” in line 214 right? NH4+?

NH3+ is right. Our monomer M4 (Figure 1) and our resin R4 (Scheme 1) contain NH3+ groups. Please, see the structures in Figure 1 and Scheme 1. Anyway, to avoid confusion we have standardized all amine groups to NH3+.

  1. “After 90 min.” in line 350 should be revised as “After 90 min,”. “.” is missing for the sentence “The conditions applied afforded very good conversions (98%)” in line 354. Please double check the whole manuscript to remove such typos.

We thank a lot the Reviewer for his observations. The issues signalled by the Reviewer have been solved. Please, see lines 363 and 367 (revised version). Additionally, as suggested by the Reviewer, the whole manuscript has been double checked with the help of Professor Deirdre Kantz, native English teacher who currently works for the University of Genoa and Pavia, to find and correct other typos and improve the English quality of our work.

  1. The scale bar in Figure 5 is hard to be recognized. Please replace it with a clear one.

According to the Reviewer request, the scale bar in Figure 5 has been replaced with a clearer one. Now the scale bar is recognizable.

  1. Please redraw the fitting curve in Figure 7b. The data before 1.75 h show good linearship.

We apologise in advance with the Reviewer, but his request cannot be satisfied, for two reasons. First, the numerical data on the x axes in Figure 7b are not hours, but the root squares of times reported on the x axes in Figure 7a (expressed in hours). Secondly, we cannot redraw the fitting curve in Figure 7b as he suggested, using only data that show good linearity (data before 1.75), because it is not the correct method for assessing what mathematical kinetic model better fit the data of a cumulative release of “something” from a matrix (%) (release of water from the swollen resin, in our case). To this end, all data of the cumulative release must be used and reworked according to various mathematical models, thus obtaining various new regression graphs, of which the tendency linear lines, their equations and their R2 values must be obtained by the last square method. At this point, the regression model which will provide the higher R2 value, will be the mathematical kinetic model that best fit the release data, and whose mechanisms govern the studied release.

  1. It would be better to do some comparison on the bactericidal performances with other polymers. More references published recently are suggested to be cited. Please double check the writing of reference. Some journal names are written in full name while others are not.

As asked by the Reviewer, a comparison on the bactericidal performances with a polymer recently reported and tested as material for water decontamination has been included with the related reference (lines 610-629). Concerning the second request by the Reviewer, we make kindly note that Polymers accept both formats. Anyway, the reference list has been modified to standardize the format, using in all references the full names of journals.

Round 2

Reviewer 2 Report

The manuscript has been revised according to the comments and could be accepted.